# Facial Expressions and Self-Reported Emotions When Viewing Nature Images

**DOI:** 10.3390/ijerph191710588

**Published:** 2022-08-25

**Authors:** Marek Franěk, Jan Petružálek, Denis Šefara

**Affiliations:** Faculty of Informatics and Management, University of Hradec Králové, Rokitanského 62, 500 03 Hradec Králové, Czech Republic

**Keywords:** face reading technique, emotional facial expressions, natural environment, positive emotions

## Abstract

Many studies have demonstrated that exposure to simulated natural scenes has positive effects on emotions and reduces stress. In the present study, we investigated emotional facial expressions while viewing images of various types of natural environments. Both automated facial expression analysis by iMotions’ AFFDEX 8.1 software (iMotions, Copenhagen, Denmark) and self-reported emotions were analyzed. Attractive and unattractive natural images were used, representing either open or closed natural environments. The goal was to further understand the actual features and characteristics of natural scenes that could positively affect emotional states and to evaluate face reading technology to measure such effects. It was predicted that attractive natural scenes would evoke significantly higher levels of positive emotions than unattractive scenes. The results showed generally small values of emotional facial expressions while observing the images. The facial expression of joy was significantly higher than that of other registered emotions. Contrary to predictions, there was no difference between facial emotions while viewing attractive and unattractive scenes. However, the self-reported emotions evoked by the images showed significantly larger differences between specific categories of images in accordance with the predictions. The differences between the registered emotional facial expressions and self-reported emotions suggested that the participants more likely described images in terms of common stereotypes linked with the beauty of natural environments. This result might be an important finding for further methodological considerations.

## 1. Introduction

A large body of research has consistently documented that people react to viewing natural environments with positive emotions. Moreover, research has also showed that exposure to simulated, surrogate nature has similar positive effects [1]. For various reasons, many people do not often have the opportunity to spend free time in natural environments. Thus, they may wish to relax for a short time during work, where some form of nature, even in a virtual medium, may help substitute for the experience of an actual natural environment. Therefore, it is useful to expand our knowledge of emotional reactions to simulated nature along with its restorative possibilities. This research employed various methods, from analyses of self-reported assessments of perceived emotions to measurements of various physiological reactions. This study analyzed changes in the emotional facial expression using machine vision software for an automated facial expression analysis of images that represent various types of natural environments and compared the results with the self-reported emotions. 

### 1.1. Positive Effects of Viewing Surrogate Nature

Over several decades, research has documented the health-improving and restorative effects of direct contact with the natural environment (for a review, see [2,3,4]). Research has also solved the question of whether exposure to simulated natural scenes may also have similar positive effects. It was determined that viewing natural images can improve mood and the level of perceived restoration [5,6,7,8,9]. Exposure to videos with a natural environment can also improve mood and the level of perceived restoration while also reducing stress (e.g., [10,11,12,13,14,15]). Recent research reported that similar effects were revealed from viewing nature scenes in virtual reality (e.g., [16,17]).

Various research methods were used in these investigations. The most frequently used self-reported assessments were employed (e.g., [5,6,8,10,18,19]). Other studies used physiological measures or combined self-reports with physiological measures such as electromyography, EEG, blood pressure, blood volume pulse, and heart rate (e.g., [10,11,12,14,20,21]). However, to date, analysis of emotional facial expression has rarely been used [22,23,24,25,26].

### 1.2. Emotional Facial Expressions

The roots of facial emotion research were established in the 19th century by Darwin [27], who argued that all humans show emotion in the face and body through remarkably similar behaviors. Darwin also conducted the first detailed study of the muscle actions involved in emotions. Within the evolutionist framework, the next step in the research of facial emotional expressions was carried out by Ekman [28] Through a series of investigations, he found high agreement across members of diverse Western and Eastern cultures on selecting emotional labels that fit facial expressions. Ekman [28] defined the six basic emotions that should be common in all cultures: anger, disgust, fear, happiness, sadness, and surprise. He believed that they can be easily recognized in facial expressions. This concept of six basic emotions has preferentially been used in analyses of facial expressions of emotions. However, more recent research questioned the cultural universality hypothesis by showing that Easterners in contrast to Westerners represent emotional intensity with distinctive dynamic eye activity (e.g., [29]).

EEG and brain imagining studies demonstrate brain specificity in the judgment of discrete emotions. The perception of fearful faces activates the region of the left amygdala (e.g., [30]) and the perception of sad faces activates the left amygdala and right temporal lobe (e.g., [31]). The perception of angry faces results in activation of the right orbito-frontal cortex and cingulate cortex (e.g., [32]). The perception of disgusted faces activates the basal ganglia, anterior insula, and frontal lobes (e.g., [31]). Duchene smiles activate the left side of the lateral frontal, midfrontal, anterior temporal, and central anterior scalp regions (e.g., [33]).

A further important point is the coherence between emotion and facial expression. The research of facial emotional expressions implicitly assumes that facial expressions co-occur with emotion. Recent research evidence (for a summary, see [34]) suggests high coherence between amusement and smiling and low to moderate coherence between other positive emotions and smiling. Surprise and disgust are accompanied by their “traditional” facial expressions. However, evidence concerning sadness, anger, and fear is still limited. For sadness, it seems that high emotion–expression coherence may exist in specific situations, whereas for anger and fear, the evidence points to low coherence. Thus, further research in this field is still needed. Some authors [35] suggest an urgent need for research that examines how people move their faces to express emotions in the variety of contexts that make up everyday life. Thus, facial emotional expressions in reaction to natural scenes is the context that has been investigated only rarely.

### 1.3. Measurement of Emotional Facial Expressions

Apart from the facial action coding system based on subjective identification of six basic emotions in video-recorded faces [28], two other methods are used: facial electromyography and automatic computer facial expression analysis. Facial electromyography is based on monitoring the activation of facial muscles during changes in the emotional response. It requires the application of electrodes on the skin surface. It enables the identification of the specific facial muscle patterns used to display emotional experiences, such as, joy, appetite, and disgust (e.g., [36]). This technique allows for the detection of subtle facial muscle activity, but its disadvantage is its technical complexity. Moreover, having electrodes attached to the face is far from naturalistic conditions.

In contrast, automated facial expression analysis by machine vision software eliminates these disadvantages [37]. These techniques have improved considerably, and their results are comparable to technically difficult facial electromyography (e.g., [38,39]). To date, automated facial expression recognition has rarely been used in environmental psychology research. In a series of experiments conducted by Wei and his colleagues [25,26,40,41], participants were asked to take selfies or were photographed while walking on urban streets or in a forest park. The photographs were analyzed using FireFACE software. In general, it was shown that compared to the urban environment, the forest experience evoked higher happiness scores but lower neutrality scores.

#### AFFDEX Software for Automatic Computer Facial Expression Analysis

The current study employed a new iMotions’ AFFDEX commercial software designed for the recognition of facial emotions [42]. The software is based on frame-to frame analysis of static images or videos. Typically, it is possible to achieve frame rates of 30 frames per second on laptop/desktop devices. It works in three steps: face detection, facial landmark detection and registration, and facial expression and emotion classification. In the first step, the position of a face is found in an image (it uses the same technology as iPhone or Android smartphones, for instance). In the next step, facial landmarks such as eyes and eye corners, brows, mouth corners, the nose tip, etc. are detected. After this, an internal face model is created. The face model is a simplified version of the respondent’s actual face; however, it contains exactly the face features for the job to be completed. Exemplary features are single landmark points (eyebrow corners, mouth corners, nose tip) and feature groups (the entire mouth, the entire arch of the eyebrows, etc.). Finally, once the simplified face model is available, the position and orientation information of all the key features is fed as input into classification algorithms, which translate the features into specific action units (specific movements of facial muscles). The recognition of emotion expressions (anger, disgust, fear, joy, sadness, surprise, and contempt) is based on combinations of these facial actions. This coding was built on Ekman’s emotional facial action coding system [28]. Automatic facial expression analysis generates numeric scores for facial expressions, action units, and emotions along with the degree of confidence. As the facial expression or emotion occurs and/or intensifies, the confidence score rises from 0 (no expression) to 100 (expression fully present). The software was tested in various preliminary explorations conducted by the Affectiva company [43], and in several independent research studies [44,45,46,47]. These studies seem to confirm that the software is reliable for recognizing basic, subtle emotional facial expressions for standardized images when participants do not intend to conceal their facial reactions; notably, the software demonstrates similar precision as the facial action coding system and facial electromyography.

### 1.4. The Current Study

In our previous study [48], facial expressions while viewing forest trees with foliage, forest trees without foliage, and urban images by iMotions’ AFFDEX software were analyzed under laboratory conditions. Although it was assumed that natural images would evoke a higher magnitude of positive emotions in facial expressions and a lower magnitude of negative emotions than urban images, the results showed very low magnitudes of emotional facial responses; moreover, the differences within both types of natural images were not significant, nor were they significant between natural and urban images. This was explained by the fact that the images represented an ordinary deciduous forest and urban streets while in the main body of research, mostly attractive natural environments were employed. This is consistent with Joye’s and Bolderdijk’s findings [49] in that they observed that watching awesome natural scenes had pronounced emotional effects compared to viewing mundane natural scenes, which generated a low emotional effect.

To further explore the possibilities of automated emotional facial expression recognition in environmental psychology research, the present study employed both attractive and unattractive “mundane” natural scenes and explored their effect on facial expressions. The measurement of facial expressions conducted by iMotions’ AFFDEX software designed for the recognition of facial emotions was combined with a self-reported description of emotions evoked by the images. Moreover, the effects of viewing open and closed scenes were explored. It was found that closed natural scenes may evoke the perception of danger and fear (e.g., [50,51,52]) in contrast to openness, which may promote visibility, a predictor of security in prospect-refuge theory [53]. This effect is more pronounced in females (e.g., [50]) because they are more afraid of potential crime than males. On the other hand, some studies showed that females may express more positive emotional expressions in natural environments than males (e.g., [51,54]).

To summarize, the goals of the present study were to further understand the actual features and characteristics of natural scenes that could positively affect emotional states and to evaluate face reading technology to measure such effects. We predicted that attractive natural scenes may evoke significantly higher levels of positive emotions than unattractive scenes, and open scenes may evoke significantly higher levels of positive emotions than closed scenes. Moreover, we predicted consistency between emotional facial expressions and self-reported emotions.

## 2. Materials and Methods

### 2.1. Participants

Fifty-one undergraduates participated in the experiment. The sample comprised young adults between the ages of 19 and 25 (mean = 20.9, SD = 1.28, 29 males, 22 females). Participants were enrolled in the first, second, or third year of various psychology courses, and they were students in informatics, financial management, or tourism at the University of Hradec Králové. Power analysis was performed using the G*Power software Version 3.1.9.7 (University of Kiel, Kiel, Germany) [55]. This study was designed to be sensitive to the detection of medium-sized effects in accordance with prior research examining the effects of attractive and unattractive nature [49]. The analysis revealed that a sample size of 34 participants would be sufficient to find significant differences (effect size f = 0.25, α = 0.05, statistical power = 0.80).

### 2.2. Stimulus Material

Twenty images were presented in one experimental session. They consisted of five images of attractive and open natural environments, five images of attractive and closed natural environments, five images of unattractive and open natural environments, and five images of unattractive and natural closed environments. In determining the differences between attractive and unattractive images, we followed the procedure used in Joy and Bolderdijk’s study [49]. They collected pictures of attractive and awesome nature from the internet that consisted of pictures of grand and dramatic mountain scenes while images of mundane nature taken by the author consisted of photographs of everyday natural elements to eliminate any “powerful” natural elements that might trigger awe. Similarly, in our study, the attractive images used were found on the Pixabay internet server, which shares copyright-free images (https://pixabay.com/ (accessed on 1 November 2020)), and unattractive images were taken by one of the authors (Figure 1). The attractive open images were scenes from high mountains, lakes, or a coast, which were fundamentally different from the common landscape of the Czech Republic, where the participants were living. The unattractive open scenes were scenes of the common Czech landscape. Attractive closed scenes were also taken from outside the Czech Republic, and downloaded from the Pixabay internet server while unattractive closed scenes were obtained from the common environment of Czech forests. The attractive scenes were professional photographs taken with the aim of having the greatest possible visual effect, given the choice of lighting, specific atmospheric conditions, and further digital adjustments, while the photos of unattractive scenes were taken in summer on mild-sunny days under normal lighting conditions, without any further adjustment. The intention was to capture how the scenes appeared in common everyday conditions. Although the perception of attractivity of natural scenes may be to some extent culturally dependent, our sample was culturally homogenous (all participants were born and living in the northeast Czechia), thus one may suppose that the selected images have a similar meaning in terms of attractiveness/unattractiveness and openness/closedness.

### 2.3. Apparatus

The experiment was controlled by a PC computer with a 1920 × 1200 pixel resolution screen and a diagonal of 61 cm with a Logitech Webcam C920 camera situated on the top of the screen. The camera and presentation of stimuli and the data processing were controlled by the iMotions 8.1 software (iMotions, Copenhagen, Denmark). The facial expression analysis was performed by the iMotions Facial Expression Analysis Module AFFDEX. The web camera recorded facial videos while the participants viewed the stimuli, and then the videos were imported into the iMotions software for facial expression analysis postprocessing. AFFDEX enables the measurement of seven emotional categories: joy, anger, surprise, fear, contempt, sadness, and disgust. Moreover, AFFDEX also provides measurement of the involvement indicators: engagement and valence. All emotional indicators were scored by the software on a scale from 0 to 100, indicating the probability of having detected the emotion. A magnitude of 0 indicated that the emotion was absent; in turn, a magnitude of 100 indicated a 100% probability of having detected the emotion.

### 2.4. Procedure

The experimental session consisted of two parts. In the first part, the participants viewed the images while the facial emotional movements were recorded. In the second part of the experiment, the participants viewed the same images and were asked to report emotions evoked by the particular images.

After arrival, the participants signed the informed consent form. Next, they underwent the first part of the experiment. They began by reading the instructions for the first part of the experiment, which is as follows: “You will see 20 images during this experiment. The images show certain natural environments. Try to imagine that you are in this natural environment right now and look closely at the picture. Each image will be displayed for 15 s. During the time you are viewing the image, the camera monitors your face.” The images were presented in a random order. Every trial started with a fixation cross situated in the center of the screen on a gray background. The participants had to fixate on the fixation cross for 2 s before the image appeared. Each image was displayed for 15 s.

After the participants completed the first part of the experiment, they began the second part. The following instruction appeared on the computer screen: “You will see the pictures you saw during the previous research session again. Take a look at them again, try to imagine that you are in this natural environment right now, and describe your perceptions and feelings by the statements, which will be given under each picture”.

### 2.5. Measures

AFFDEX registered the probability of having detected facial emotions related to joy, anger, surprise, fear, contempt, sadness, and disgust and evaluated the involvement indicators engagement (the emotional responsiveness that the stimuli trigger) and valence (the positive or negative nature of the experience). A questionnaire used in the second part of the experiment registered four basic emotions, specifically joy, surprise, fear, and sadness. We did not ask about anger and contempt because we assumed that natural environments would not provoke these negative emotions. The item “I feel happy here” was related to the emotion joy, the item “I feel amazed here” was related to the emotion surprise, the item “This place scares me a little” was related to the emotion fear, and finally, the item “This is a pretty sad place” was related to sadness. The item “I like this place” was related to liking the environment. Each item was assessed on a five-point scale (1 = not at all, 5 = completely). The questionnaire was presented to the participants on a computer screen.

## 3. Results

AFFDEX offers various outputs of data. One possibility, which is suitable for the purposes of the current research, is to work with raw data that express the intensity of the facial emotions or the involvement indicators on a scale from 0 to 100 found in approximately 30-ms time windows. Thus, firstly, the raw data were exported. Within the measurement frequency of 30 frames per second, approximately 450 measurements were obtained for one image presented for 15 s, and approximately 2250 measurements were obtained for one participant within one image category (attractive open images, attractive closed images, unattractive open images, unattractive closed images). Next, the mean scores of the intensities of specific emotions that appeared were calculated for each participant regarding the images in each category and the involvement indicators; both sets of results were then averaged across the image categories (Table 1 and Figure 2).

### 3.1. Analysis of Emotional Facial Expressions

Three-way mixed ANOVAs were conducted to assess the effects of attractiveness, openness, and gender on emotional facial expressions. Attractiveness (attractive × unattractive), openness (open × closed), and gender (male × female) were chosen as predictors, and the emotional facial expression or involvement indicator was chosen as a dependent variable.

**Anger.** There was no statistically significant three-way interaction between attractiveness, openness, and gender (*F* (1, 49) = 1.126, *p* = 0.294, partial η^2^ = 0.022). There was no statistically significant two-way interaction between attractiveness and gender (*F* (1, 49) = 0.671, *p* = 0.417, partial η^2^ = 0.014), no statistically significant two-way interaction between openness and gender (*F* (1, 49) = 0.716, *p* = 0.402, partial η^2^ = 0.014), and no statistically significant two-way interaction between attractiveness and openness (*F* (1, 49) = 0.000, *p* = 0.993, partial η^2^ = 0.000). The main effects of attractiveness (*F* (1, 49) =1.151, *p* = 0.224, partial η^2^ = 0.030) and openness (*F* (1, 49) = 0.283, *p* = 0.597, partial η^2^ = 0.006) were not significant.

**Disgust.** There was no statistically significant three-way interaction between attractiveness, openness, and gender (*F* (1, 49) = 0.425, *p* = 0.518, partial η^2^ = 0.009). There was no statistically significant two-way interaction between attractiveness and gender (*F* (1, 49) = 0.462, *p* = 0.616, partial η^2^ = 0.005), no statistically significant two-way interaction between openness and gender (*F* (1, 49) = 0.712, *p* = 0.403, partial η^2^ = 0.014), and no statistically significant two-way interaction between attractiveness and openness (*F* (1, 49) = 0.862, *p* = 0.368, partial η^2^ = 0.017). The main effects of attractiveness (*F* (1, 49) = 0.255, *p* = 0.616, partial η^2^ = 0.005) and openness (*F* (1, 49) = 1.113, *p* = 0.292, partial η^2^ = 0.023) were not significant.

**Sadness.** There was no statistically significant three-way interaction between attractiveness, openness, and gender (*F* (1, 49) = 3.407, *p* = 0.071, partial η^2^ = 0.065). There was no statistically significant two-way interaction between attractiveness and gender (*F* (1, 49) = 0.551, *p* = 0.462, partial η^2^ = 0.011), no statistically significant two-way interaction between openness and gender (*F* (1, 49) = 1.036, *p* = 0.314, partial η^2^ = 0.021), and no statistically significant two-way interaction between attractiveness and openness (*F* (1, 49) = 0.019, *p* = 0.890, partial η^2^ = 0.000). The main effects of attractiveness (*F* (1, 49) = 0.887, *p* = 0.351, partial η^2^ = 0.018) and openness (*F* (1, 49) = 1.237, *p* = 0.272, partial η^2^ = 0.025) were not significant.

**Fear.** There was no statistically significant three-way interaction between attractiveness, openness, and gender (*F* (1, 49) = 1.589, *p* = 0.214, partial η^2^ = 0.031). There was no statistically significant two-way interaction between attractiveness and gender (*F* (1, 49) = 0.016, *p* = 0.899, partial η^2^ = 0.000), no statistically significant two-way interaction between openness and gender (*F* (1, 49) = 0.405, *p* = 0.528, partial η^2^ = 0.008), and no statistically significant two-way interaction between attractiveness and openness (*F* (1, 49) = 0.000, *p* = 0.990, partial η^2^ = 0.000). The main effects of attractiveness (*F* (1, 49) = 1.512, *p* = 0.225, partial η^2^ = 0.030) and openness (*F* (1, 49) = 1.274, *p* = 0.264, partial η^2^ = 0.025) were not significant.

**Contempt**. The effects of attractiveness (*F* (1, 49) = 0.344, *p* = 0.558), openness (*F* (1, 49) = 0.721, *p* = 0.397), and gender (*F* (1, 49) = 2.378, *p* = 0.890) were nonsignificant. There were no significant interactions. There was no statistically significant three-way interaction between attractiveness, openness, and gender (*F* (1, 49) = 0.138, *p* = 0.659, partial η^2^ = 0.003). There was no statistically significant two-way interaction between attractiveness and gender (*F* (1, 49) = 0.216, *p* = 0.644, partial η^2^ = 0.004), no statistically significant two-way interaction between openness and gender (*F* (1, 49) = 1.159, *p* = 0.168, partial η^2^ = 0.038), and no statistically significant two-way interaction between attractiveness and openness (*F* (1, 49) = 0.197, *p* = 0.659, partial η^2^ = 0.004). The main effects of attractiveness (*F* (1, 49) = 0.280, *p* = 0.599, partial η^2^ = 0.006) and openness (*F* (1, 49) = 1.170, *p* = 0.190, partial η^2^ = 0.035) were not significant.

**Joy.** There was no statistically significant three-way interaction between attractiveness, openness, and gender (*F* (1, 49) = 0.010, *p* = 0.921, partial η^2^ = 0.000). There was no statistically significant two-way interaction between attractiveness and gender (*F* (1, 49) = 0.533, *p* = 0.469, partial η^2^ = 0.011), no statistically significant two-way interaction between openness and gender (*F* (1, 49) = 0.487, *p* = 0.488, partial η^2^ = 0.010), and no statistically significant two-way interaction between attractiveness and openness (*F* (1, 49) = 0.062, *p* = 0.804, partial η^2^ = 0.001). The main effects of attractiveness (*F* (1, 49) = 0.127, *p* = 0.723, partial η^2^ = 0.003) and openness (*F* (1, 49) = 2.648, *p* = 0.110, partial η^2^ = 0.051) were not significant. 

**Surprise.** There was no statistically significant three-way interaction between attractiveness, openness, and gender (*F* (1, 49) = 1.046, *p* = 0.311, partial η^2^ = 0.021). There was no statistically significant two-way interaction between attractiveness and gender (*F* (1, 49) = 0.533, *p* = 0.355, partial η^2^ = 0.018), no statistically significant two-way interaction between openness and gender (*F* (1, 49) = 0.203, *p* = 0.655, partial η^2^ = 0.015), and no statistically significant two-way interaction between attractiveness and openness (*F* (1, 49) = 0.930, *p* = 0.340, partial η^2^ = 0.019). The main effects of attractiveness (*F* (1, 49) = 1.538, *p* = 0.221, partial η^2^ = 0.030) and openness (*F* (1, 49) = 0.731, *p* = 0.397, partial η^2^ = 0.015) were not significant.

**Valence.** There was no statistically significant three-way interaction between attractiveness, openness, and gender (*F* (1, 49) = 0.167, *p* = 0.684, partial η^2^ = 0.003). There was no statistically significant two-way interaction between attractiveness and gender (*F* (1, 49) = 0.155, *p* = 0.889, partial η^2^ = 0.003), no statistically significant two-way interaction between openness and gender (*F* (1, 49) = 0.827, *p* = 0.368, partial η^2^ = 0.017), and no statistically significant two-way interaction between attractiveness and openness (*F* (1, 49) = 0.042, *p* = 0.839, partial η^2^ = 0.001). The main effects of attractiveness (*F* (1, 49) = 0.187, *p* = 0.667, partial η^2^ = 0.004) and openness (*F* (1, 49) = 2.689, *p* = 0.107, partial η^2^ = 0.052) were not significant.

**Engagement.** There was no statistically significant three-way interaction between attractiveness, openness, and gender (*F* (1, 49) = 0.335, *p* = 0.566, partial η^2^ = 0.007). There was no statistically significant two-way interaction between attractiveness and gender (*F* (1, 49) = 0.155, *p* = 0.696, partial η^2^ = 0.003), no statistically significant two-way interaction between openness and gender (*F* (1, 49) = 0.356, *p* = 0.554, partial η^2^ = 0.007), and no statistically significant two-way interaction between attractiveness and openness (*F* (1, 49) = 0.408, *p* = 0.990, partial η^2^ = 0.008). The main effects of attractiveness (*F* (1, 49) = 0.040, *p* = 0.842, partial η^2^ = 0.001) and openness (*F* (1, 49) = 0.457, *p* = 0.502, partial η^2^ = 0.009) were not significant.

### 3.2. Analysis of Self-Reported Emotions

The mean scores of the self-reported emotions of joy, surprise, fear, and sadness were calculated for each category of images (Table 2 and Figure 3). Three-way mixed ANOVAs were conducted to assess the effects of attractiveness, openness, and gender on perceived emotion. Attractiveness (attractive × unattractive), openness (open × closed), and gender (male × female) were chosen as predictors, and the level of self-reported emotion was chosen as a dependent variable.

**Joy.** There was no statistically significant three-way interaction between attractiveness, openness, and gender (*F* (1, 47) = 0.553, *p* = 0.461, partial η^2^ = 0.012). Moreover, there was no statistically significant two-way interaction between attractiveness and gender (*F* (1, 47) = 0.019, *p* = 0.892, partial η^2^ = 0.000) and no statistically significant two-way interaction between openness and gender (*F* (1, 47) = 1.107, *p* = 0.298, partial η^2^ = 0.023). However, there was a statistically significant two-way interaction between attractiveness and openness (*F* (1, 47) = 102.728, *p* = 0.001, partial η^2^ = 0.686). In the attractive environments, there was a statistically significant difference in self-reported joy between open (3.824 ± 0.160 evaluation points) and non-open (3.171 ± 0.175 evaluation points) environments (*F* (1, 48) = 118.541, *p* = 0.001, partial η^2^ = 0.712), a mean difference of 0.635 (95% CI, 0.532 to 0.774) self-reported joy evaluation points. In the non-attractive environments, self-reported joy was not statistically significantly different in the open environments (2.771 ± 0.207 joy evaluation points) compared to non-open environments (2.776 ± 0.200 joy evaluation points) in the non-attractive environments (*F* (1, 48) = 0.061, *p* = 0.806, partial η^2^ = 0.001), a difference of 0.04 (95% CI, −0.037 to 0.029) self-reported joy evaluation points. Thus, openness and attractiveness have a combined effect, where attractive and open environments evoked a higher self-reported joy than attractive closed environments. In the non-attractive environments, the openness/closedness of the environments had no effect on self-reported joy.

**Surprise.** There was no statistically significant three-way interaction between attractiveness, openness, and gender (*F* (1, 47) = 0.131, *p* = 0.719, partial η^2^ = 0.003). Moreover, there was no statistically significant two-way interaction between attractiveness and gender (*F* (1, 47) = 0.388, *p* = 0.536, partial η^2^ = 0.008) and no statistically significant two-way interaction between openness and gender (*F* (1, 47) = 0.005, *p* = 0.941, partial η^2^ = 0.000). However, there was a statistically significant two-way interaction between attractiveness and openness (*F* (1, 47) = 74.750, *p* = 0.001, partial η^2^ = 0.614). In the attractive environments, there was a statistically significant difference in self-reported surprise between open (4.000 ± 0.157 evaluation points) and non-open environments (3.176 ± 0.182 evaluation points) (*F* (1, 48) = 76.681, *p* = 0.001, partial η^2^ = 0.615), a mean difference of 0.824 (95% CI, 0.635 to 1.014) self-reported surprise evaluation points. In contrast, in the non-attractive environments, self-reported surprise was not statistically significantly different in the open (2.204 ± 0.215 surprise evaluation points) compared to non-open (2.230 ± 0.216 surprise evaluation points) environments (*F* (1, 48) = 2.408, *p* = 0.127, partial η^2^ = 0.048), a difference of 0.26 (95% CI, −0.008 to 0.059) self-reported surprise emotion evaluation points. Thus, openness and attractiveness have a combined effect, where attractive and open environments evoked higher self-reported surprise than attractive closed environments. In the non-attractive environments, the openness/closedness of the environments had no effect on self-reported surprise.

**Fear.** There was no statistically significant three-way interaction between attractiveness, openness, and gender (*F* (1, 47) = 3.550, *p* = 0.066, partial η^2^ = 0.070) and no statistically significant two-way interaction between attractiveness and gender (*F* (1, 47) = 1.739, *p* = 0.194, partial η^2^ = 0.036). However, there was a statistically significant two-way interaction between openness and gender (*F* (1, 47) = 9.955, *p* = 0.003, partial η^2^ = 0.175) and a statistically significant two-way interaction between attractiveness and openness (*F* (1, 47) = 59.482, *p* = 0.001, partial η^2^ = 0.559). In the attractive environments, there was a statistically significant difference in self-reported fear between open (1.531 ± 0.141 evaluation points) and non-open environments (2.212 ± 0.249 evaluation points) that were attractive (*F* (1, 48) = 53.168, *p* = 0.001, partial η^2^ = 0.526), a mean difference of 0.682 (95% CI, 0.494 to 0.870) self-reported fear evaluation points. In contrast, in the non-attractive environments, self-reported fear was not statistically significantly different in open (1.384 ± 0.146 fear evaluation points) compared to non-open (1.342 ± 0.132 fear evaluation points) environments (*F* (1, 48) = 5.513, *p* = 0.023, partial η^2^ = 0.103), a difference of 0.42 (95% CI, 0.006 to 0.078) self-reported fear evaluation points. In closed environments, there was a statistically significant difference in self-reported fear between males (1.603 ± 0.214 evaluation points) and females (2.029 ± 0.259 evaluation points) (*F* (1, 47) = 6.481, *p* = 0.014, partial η^2^ = 0.121), a mean difference of 0.425 (95% CI, 0.089 to 0.761) self-reported fear evaluation points. In contrast, in open environments, self-reported fear was not statistically significantly different among males (1.393 ± 0.214 fear evaluation points) compared to females (1.550 ± 0.199 fear evaluation points) (*F* (1, 47) = 1.487, *p* = 0.229, partial η^2^ = 0.031), a difference of 0.157 (95% CI, −0.102 to 0.416) self-reported fear emotion evaluation points. Thus, openness and attractiveness have a combined effect, where attractive and closed environments evoked a higher self-reported fear than attractive open environments. In the non-attractive environments, the openness/closedness of the environments had no effect on self-reported fear. Moreover, in closed environments, females reported higher fear than in open environments.

**Sadness**. There was no statistically significant three-way interaction between attractiveness, openness, and gender (*F* (1, 47) = 0.528, *p* = 0.471, partial η^2^ = 0.011), no statistically significant two-way interaction between attractiveness and gender (*F* (1, 47) = 0.1399, *p* = 0.243, partial η^2^ = 0.029), and no statistically significant two-way interaction between openness and gender (*F* (1, 47) = 1.361, *p* = 0.249, partial η^2^ = 0.028). However, there was a statistically significant two-way interaction between attractiveness and openness (*F* (1, 47) = 57.565, *p* = 0.001, partial η^2^ = 0.551). In attractive environments, there was a statistically significant difference in self-reported sadness emotion between open (1.506 ± 0.137 evaluation points) and non-open (2.286 ± 0.250 evaluation points) environments (*F* (1, 48) = 65.633, *p* = 0.001, partial η^2^ = 0.578), a mean difference of 0.780 (95% CI, 0.586 to 0.973) self-reported sadness evaluation points. In contrast, in non-attractive environments, self-reported sadness was not statistically significantly different in the open (1.641 ± 0.178 sadness evaluation points) compared to non-open (1.622 ± 0.177 sadness evaluation points) (*F* (1, 48) = 0.739, *p* = 0.394, partial η^2^ = 0.015) environments, a difference of 0.018 (95% CI, −0.025 to 0.061) self-reported sadness evaluation points. Thus, openness and attractiveness have a combined effect, where attractive and closed environments evoked a higher self-reported sadness than attractive open environments. In the non-attractive environments, the openness/closedness of the environments had no effect on self-reported sadness.

## 4. Discussion

The present study explored whether attractive natural scenes evoke more positive facial emotional expressions than unattractive scenes and whether self-reported emotions are linked with objective measures of emotional facial expressions. AFFDEX software was used for automatic registrations of emotional facial expressions. Our results showed generally small magnitudes of emotional facial expressions while observing the images. The facial expression of joy was higher than that of the other registered emotions. Clearly, it is not surprising that natural environments evoke positive rather than negative emotions. However, contrary to our assumptions, there was no difference between facial emotions while viewing attractive and unattractive scenes, and between open and closed scenes. Thus, even attractive natural images did not generally evoke strong immediate emotional reactions.

To date, there is a lack of data from studies within the field of environmental psychology using a similar research methodology. Thus, we compared our results with data from recent studies that was conducted in the field of advertisement assessment research. In the study by Otamendi and Sutil [56], the participants viewed an advertisement lasting 91 s that consisted of 31 scenes. The advertisement showed the accompanying role that a mother plays throughout the life of a child, from birth to adulthood. The data was processed by the same AFFDEX software that was used in our study. The authors also reported small values for specific emotions, with the highest for joy with a mean = 4.82, and smaller for the other emotions, with means between 0.42 and 1.12. Only in the target group (mature aged women) did higher emotional reactions occur (mean for joy = 14.17). A further study [57] explored the emotional reaction evoked by two types of drinks. For instance, the mean magnitudes for the expressions for joy were 0.56 or 0.97, respectively. Even at such low magnitudes, the differences were statistically significant. Furthermore, in our previous study [48], where ordinary natural scenes were used, very small mean magnitudes ranging from 0.10 to 0.14 for the emotional expression of joy in viewing natural images were registered while the mean values of joy ranged from approximately 1.20–1.80 in individual participants in the present study. However, this increase in the facial expression appearance of joy was probably not caused by the effect of the attractiveness of images because the mean values of joy for attractive and unattractive images were almost equal. It is supposed that the increase in positive facial emotions was due to the instruction given to the participants, where we asked them not only to view the images as in the previous study [48] but also encouraged the participants to try and imagine that they were immersed in the specific environment.

One explanation for the very low magnitude level of facial expressions produced during the observation of natural stimuli could be based on the fact that the software for automated facial expression analysis cannot integrate contextual information into emotion recognition. Given that from an evolutionary perspective, facial expressions have an intrinsic communicative function [27,28], there might be a difference in the emotional facial expression of individuals sitting alone in a laboratory and individuals who want to share their emotional experience with someone else. It seems that the studies by Wei and his colleagues [25,26,40,41] identified significant differences between emotional facial expressions in natural and urban environments because they analyzed facial expressions on selfies that are usually taken to communicate our emotional experience on a given place and situation with our friends.

In contrast to registered emotional facial expressions, the self-reported emotions evoked by the images showed significantly larger differences between specific categories of images. As predicted, the attractive images evoked a higher level of positive emotions than unattractive images. Conversely, unattractive images evoked higher levels of negative emotions than attractive images. Importantly, we found a combined effect between the attractiveness of images and their openness; specifically, attractive and open environments evoked higher self-reported joy and surprise than attractive closed environments. Conversely, attractive and closed environments evoked higher self-reported fear and sadness than attractive open environments. Moreover, in closed attractive environments, females reported higher fear than in open attractive environments, which is in accordance with a large body of environmental psychology research showing that females may fear being attacked in closed natural environments (e.g., [51,54]).

Thus, the results showed that there was a difference between emotional facial expressions directly evoked by images viewed and subjective statements about emotions evoked by the same images. On the one hand, strong self-reported emotions might be given just by the fact that we are accustomed to describing natural environments in a positive way. Thus, it might be that the participants did not fully describe the intensity of their actual emotions; rather, they more likely described images in terms of common stereotypes linked with the beauty of natural environments. This might be an important finding for further methodological considerations.

Furthermore, this finding should be discussed in terms of differences between macro- and micro-emotional expressions [58]. People express their emotions consciously by macro-expressions that last from 0.5 to 4 s [58], and the emotions are easily recognized. Micro-expression is mostly expressed unconsciously and reflects a person’s true emotions. According to Ekman [58], the duration of micro-expressions typically lasts between 0.5 and 4 s; more recent research proposed that they are shorter, under 500 ms [59]. Our data represented means along emotional facial expressions that appeared in 15 s intervals, corresponding to when images were presented. Because eyes move across an image, actual short emotional expressions may change. Thus, a next step in this research would be to analyze whether there might be a link between certain environmental features and immediate corresponding emotional expressions.

It is also worth commenting on the appearance of the facial expressions of contempt and disgust while viewing the natural images. We did not assume that the natural images used in the experiment evoked these emotions; rather, they were linked with concentration on the task. It was shown that nose wrinkles, which are associated with disgust, were also present in situations where participants were asked to learn information presented on a computer screen [60,61] and were not always observed with self-reports of disgust [62].

The limitation of the present findings is related to the laboratory situation. Although many studies showed positive effects of real outdoor environments and their simulation in a laboratory [1], a stronger effect may be observed with self-selected natural images or in situations where people decide to view natural images by themselves to relax or to remember the nice moments they have spent in nature. Another limitation is the small number of photographs that were used in this study and their specific selection. Obviously, natural scenes take many diverse forms all over the world, and a single investigation cannot include them all. Thus, the present study does not claim to be generalizable. Further replication studies are needed.

## 5. Conclusions

This study analyzed changes in emotional facial expression through the use of machine vision software of images representing various types of natural environments and compared the results with the self-reported emotions. While self-reported emotions reflected differences between the types of natural environments, there was no difference between facial emotions while viewing attractive and unattractive scenes. The results might suggest that participants might not have described the intensity of their actual emotions; rather, they more likely described images in terms of common stereotypes linked with the beauty of natural environments. The results also contribute to the considerations of the use of automated facial expression analysis in environmental psychology research and further expansion to understand facial emotional expressions in a specific situation.

## Figures and Tables

**Figure 1 ijerph-19-10588-f001:**
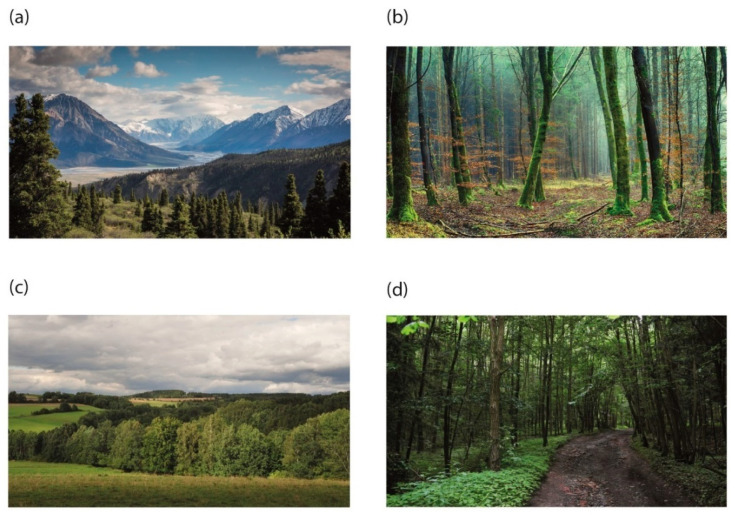
Examples of the stimulus material: (**a**)—attractive open environment, (**b**)—attractive closed environment, (**c**)—unattractive open environment, and (**d**)—unattractive closed environment.

**Figure 2 ijerph-19-10588-f002:**
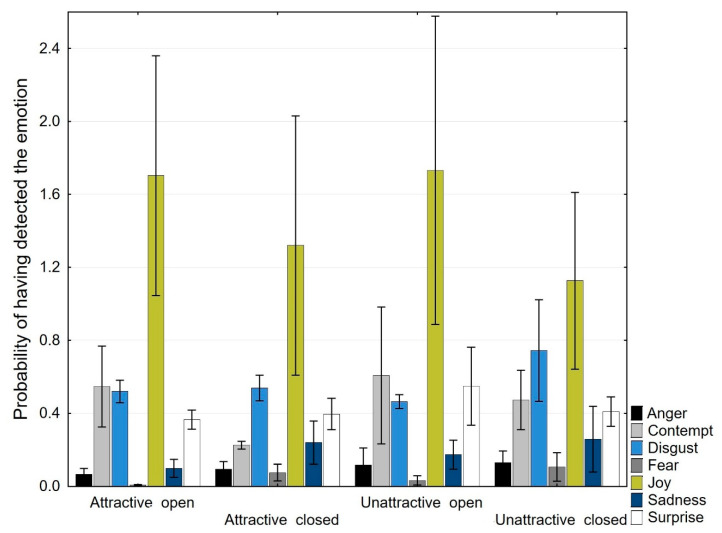
Mean scores and standard errors of the means for individual emotional categories with exposure to an attractive open environment, attractive closed environment, unattractive open environment, and unattractive closed environment (the scale ranged from 0 to 100).

**Figure 3 ijerph-19-10588-f003:**
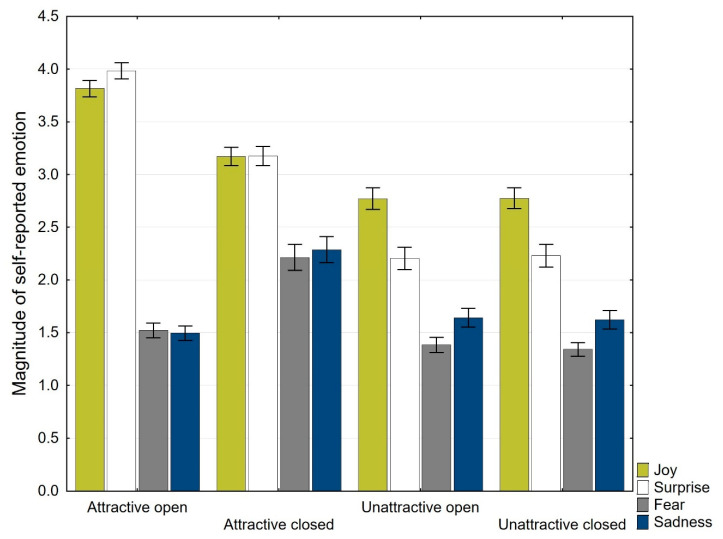
Mean scores and standard errors of the means for the self-reported emotions of joy, surprise, fear, and sadness (the scale ranged from 1 to 5).

**Table 1 ijerph-19-10588-t001:** Mean scores for individual emotional categories and involvement indicators with exposure to an attractive open environment, attractive closed environment, unattractive open environment, and unattractive closed environment (the scale ranged from 0 to 100).

	Attractive Open		Attractive Closed		Unattractive Open		Unattractive Closed	
	Mean	SD	Mean	SD	Mean	SD	Mean	SD
**Emotion**								
Anger	0.066	0.230	0.096	0.288	0.117	0.668	0.129	0.455
Contempt	0.546	1.582	0.225	0.145	0.607	2.672	0.472	1.159
Disgust	0.519	0.438	0.538	0.500	0.464	0.280	0.744	1.992
Fear	0.009	0.016	0.075	0.323	0.033	0.179	0.106	0.557
Joy	1.703	4.701	1.319	5.080	1.732	6.040	1.126	3.463
Sadness	0.098	0.347	0.239	0.836	0.173	0.564	0.258	1.281
Surprise	0.365	0.375	0.395	0.613	0.549	1.529	0.410	0.573
**Involvement indicators**								
Engagement	4.972	7.449	4.275	8.492	4.515	8.885	4.538	7.111
Valence	1.317	6.369	0.669	6.129	1.259	7.975	0.457	6.147

**Table 2 ijerph-19-10588-t002:** Mean scores for the self-reported emotions of joy, surprise, fear, and sadness (the scale ranged from 1 to 5).

	Attractive Open		Attractive Closed		Unattractive Open		Unattractive Closed	
	Mean	SD	Mean	SD	Mean	SD	Mean	SD
**Emotion**								
Joy	3.8	0.56	3.2	0.62	2.8	0.72	2.8	0.69
Surprise	4.0	0.55	3.2	0.63	2.2	0.75	2.2	0.75
Fear	1.5	0.49	2.2	0.87	1.4	0.51	1.3	0.46
Sadness	1.5	0.48	2.3	0.87	1.6	0.62	1.6	0.74

## Data Availability

The datasets supporting this article have been uploaded as part of the Appendix A.

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
