# Peer review of "Facial Expressions and Self-Reported Emotions When Viewing Nature Images"

_ijerph, 2022, doi:10.3390/ijerph191710588_

Round 1
Reviewer 1 Report
GENERAL COMMENTS
The article “Facial expressions and self-reported emotions when viewing nature images” of Franek and co-workers tries to further deepen the emotional reaction related to the observation of natural context.
In particular, the study is aimed at examining emotional reactions of the participants by means of their facial expression production and trough a brief self-reported emotion questionnaire.
I think the paper is quite well written and it could in some way help our scientific understanding of emotion processes, as well as the implications of such knowledges for clinical interventions (such as stress reduction protocols).
Unfortunately, I think the paper needs an important revision process in order to be considered for publication, especially with regard of the analyses model implemented and some important lack in the theoretical framework of the study.
MINOR AND MAJOR CONCERNS
1) ABSTRACT, INTRODUCTION
a) I think it is quite important to better clarify general goals and aims of the study both in the abstract and in the main text.
Is it to further understand what are the actual features and characteristics of natural scenes that could positively affect emotional states and to evaluate the best methodology to measure such effects?
b) I think it is important also to clarify why the chosen methodology – that is evaluating emotional facial expressions production during the observation of natural scenes – would help in addressing the goals of the study.
To this end, it is also important to stress the link between facial expressions and emotional states. Even it is not the main topics of the study, nevertheless it is a crucial link to support the aims of the study, methodology adopted and results.
Besides the classic works of Darwin (“the expression of emotion in Man and Animals”) and Ekman on the relevance of facial expressions for emotional processes, I also suggest some papers of the neuroscientific field, showing that facial expressions production/elaboration involve brain regions linked both to the control of facial expression behavioral/motor pattern production (such as cortical sensorimotor regions) as well as other structures (such as insula, cingulate cortex, amygdala) which modulates the autonomic activity typically coupled with emotion (see Gerbella et al., 2021; Caruana et al. 2018, 2017, 2011; Carr et al., 2003;). To this regard, to further support these aspects, I think it would be useful also to briefly discuss some work showing that blocking facial expressions due to artificial experimental protocols (Borgomaneri et al., 2020; Oberman et al., 2007) or because of clinical conditions, such as the facial amimia in Parkinson’s disease (Argoud et al., 2018) or congenital impossibility to produce facial expressions (see for example, Belluardo et al., 2022; de Stefani et al., 2019; Nicolini et al; 2019;), seems to negatively affect emotional process both in terms of emotion recognition and autonomic responses to emotional stimuli.
2) METHODS, EXPERIMENTAL PROCEDURE, DATA ELABORATION AND RESULTS
a) It is not clear how AFFDEX software process the video and how it obtains the probability of having detected a facial expression. What does it mean that you had 450 measurements per image? (page 5, line 186). Please clarify.
In general, to better understand the results, I think it would be helpful to give some more detailed information about the iMotion software AFFDEX functioning, in particular on the video-processing and post-processing procedure. For example, what is the sampling rate (how many images per second)? does it uses templates for each facial expressions and it compares the facial gestures recorded with these templates?
Moreover, it is important to clarify the meaning of the other outputs besides the probability of detecting facial expressions, that is what is the meaning of “valence” and “engagement” and how to interpret their mean values.
b) Improve figures: they do not well represent the results (I don’t understand if they have been uploaded wrong files, but they do not have standard errors nor markers indicating statistically significant results).
c) Data analysis
i) Analysis of facial expressions
- Why do you have 3 “involvement indicators”? (Page 5, line 199) previously in the text you wrote that the software gives 2 involvement indicators (Valence and Engagement).
- Improve the description of the analysis model used. In my opinion it is not clear if the first ANOVA was conducted not including the different category of images as predictors, so it is an overall analysis comparing the magnitude of facial expressions and involvement? In that case, I don’t understand why to perform this analysis considering that in the second ANOVA you would have overall main results as well as interactions.
- Moreover, it seems you performed several different ANOVA for each emotion and involvement variables separately. If this is the case, I don’t think this procedure is statistically correct.
In general, I think you should just have an overall model with emotional variables (emotion category and involvement ones), attractiveness and openness as within-subject factors and gender as a between factor, and then see the main and/or interaction effect. Please clarify
ii) Analysis of self-reported questionnaire
- Same concerns as for the facial expression analysis. You performed separate ANOVA for each emotion, but you should perform an overall model having the level of self-reported scores as dependent variables, with emotional categories and image characteristics as within-subject predictors and gender as a between-subject factor. Please clarify
3) DISCUSSION AND CONCLUSIONS
Just a suggestion here, that is to discuss the very low magnitude level of facial expressions production during the observation of natural stimuli also considering that from evolutionary/ethological perspective facial expressions have an intrinsic communicative value/function. In other words, being alone and not sharing with someone else the experience of observing natural scenes may limit the actual production of facial expressions
Author Response
Dear reviewer,
thank you for your comments and recommendation. We tried to address all of them. Bellow, we described our changes and modifications. Your comments are in red color, our answers are black.
1) ABSTRACT, INTRODUCTION
- a)I think it is quite important to better clarify general goals and aims of the study both in the abstract and in the main text. Is it to further understand what are the actual features and characteristics of natural scenes that could positively affect emotional states and to evaluate the best methodology to measure such effects?
We clarified the general goals and aims in the abstract and in the main text (subsection 1.4. The current study). We also emphasized methodological orientation of the study. We take liberty to borrow your sentence because it precisely describes what we wanted to do.
- b)I think it is important also to clarify why the chosen methodology – that is evaluating emotional facial expressions production during the observation of natural scenes – would help in addressing the goals of the study. To this end, it is also important to stress the link between facial expressions and emotional states. Even it is not the main topics of the study, nevertheless it is a crucial link to support the aims of the study, methodology adopted and results.
We inserted a new subsection “1.2. Emotional facial expressions”, where we briefly introduced the concept of facial emotions in connection with the works by Drawing and namely by Ekman. Next, we briefly described neurophysiological findings concerning responses to specific facial emotions in human brain. Although the phenomena linked with blocking facial emotions are interesting, we skipped this topic because it is more related to recognition of emotion on human faces, which is a bit far from the topic of our paper. Instead, we discussed important question of coherence between emotion and facial expressions.
2) METHODS, EXPERIMENTAL PROCEDURE, DATA ELABORATION AND RESULTS
- a)It is not clear how AFFDEX software process the video and how it obtains the probability of having detected a facial expression. What does it mean that you had 450 measurements per image? (page 5, line 186). Please clarify. In general, to better understand the results, I think it would be helpful to give some more detailed information about the iMotion software AFFDEX functioning, in particular on the video-processing and post-processing procedure. For example, what is the sampling rate (how many images per second)? does it uses templates for each facial expressions and it compares the facial gestures recorded with these templates? Moreover, it is important to clarify the meaning of the other outputs besides the probability of detecting facial expressions, that is what is the meaning of “valence” and “engagement” and how to interpret their mean values.
We inserted the subsection “1.3.1. AFFDEX software for automatic computer facial expression analysis“, where we described, how the software works. Further explanation is in the subsection “3. Results”, lines 300-305. The meaning of “valence” and “engagement was explained.
- b)Improve figures: they do not well represent the results (I don’t understand if they have been uploaded wrong files, but they do not have standard errors nor markers indicating statistically significant results).
Standard error bars were added in both figures, but we did not add markers indicating statistically significant results. We are afraid that due to the large number of bars in the graphs, this designation would not be very clear, so we left this information only in the text.
- c)Data analysis
- i)Analysis of facial expressions
- Why do you have 3 “involvement indicators”? (Page 5, line 199) previously in the text you wrote that the software gives 2 involvement indicators (Valence and Engagement).
Corrected. There were 2 involvement indicators.
- Moreover, it seems you performed several different ANOVA for each emotion and involvement variables separately. If this is the case, I don’t think this procedure is statistically correct. In general, I think you should just have an overall model with emotional variables (emotion category and involvement ones), attractiveness and openness as within-subject factors and gender as a between factor, and then see the main and/or interaction effect. Please clarify. Same concerns as for the facial expression analysis. You performed separate ANOVA for each emotion, but you should perform an overall model having the level of self-reported scores as dependent variables, with emotional categories and image characteristics as within-subject predictors and gender as a between-subject factor. Please clarify
I discussed your recommendation with a statistician. Despite understanding your concerns, we feel that using MANOVA would not be appropriate in our case. As MANOVA essentially combines the two or more dependent variables to form a 'new' dependent variable in such a way as to maximize the differences between the groups of the independent variables, it is expected such dependent variables have at least some level of mutual consistency/coherency (for example – when investigating school study results, it could be beneficial to make use of MANOVA where combined dependent variable is consisting of multiple different school exams grades, etc.). In our case, however, emotion categories are rather distinct (AFFDEX defines them as basic emotions), this is why it feels inappropriate for us to combine for example Joy and Disgust (or even other emotions) together to carry out and MANOVA as such emotions contradict each other.
Improve the description of the analysis model used. In my opinion it is not clear if the first ANOVA was conducted not including the different category of images as predictors, so it is an overall analysis comparing the magnitude of facial expressions and involvement? In that case, I don’t understand why to perform this analysis considering that in the second ANOVA you would have overall main results as well as interactions.
Because we consider MANOVA as inappropriate for our data, this one-way repeated-measures ANOVA is a simple way, how to confirm that magnitudes of facial emotion Joy were significantly higher that magnitudes of other emotions. Because in all conditions the magnitudes of particular emotions clearly showed almost the same pattern, we averaged data over all conditions. We considered a more complex ANOVA model unnecessary because it would show nothing more that the simple model of one-way ANOVA.
3) DISCUSSION AND CONCLUSIONS
Just a suggestion here, that is to discuss the very low magnitude level of facial expressions production during the observation of natural stimuli also considering that from evolutionary/ethological perspective facial expressions have an intrinsic communicative value/function. In other words, being alone and not sharing with someone else the experience of observing natural scenes may limit the actual production of facial expressions.
Thank you for this comment. We discussed this explanation on lines 547-557.
Reviewer 2 Report
The authors focused on an interesting topic: if and how viewing images of various types of natural environments can influence human's emotional facial expressions. Both automated facial expression analysis and self-reported emotions are compared and analyzed for the facial expression measurement. This paper is easy to follow. However, the experiment and evaluation were conducted on 51 participants and a total of 20 images with attractive/unattractive/open/closed natural environments. One major concern is that the "attractive" and "unattractive" are not well-defined. They are quite subjective experiences, which can be influenced by gender, age, education, and other personal experience. Further, the small number of tested images maybe not be sufficient and convincing to reach a definite conclusion. More detailed concerns are shown as follows.
1. This study investigated the effect of viewing attractive or unattractive natural scenes on emotional reactions. From the examples of the stimulus material shown in Figure 1, we can see that the "attractive" and "unattractive" are relative. The authors explained that "photos of unattractive scenes were taken on normal summer sunny days without any further adjustment". I don't think the adjustment is the main factor to distinguish "attractive" images from "unattractive" ones. For example, the nature scene, weather, camera angle, and even the image size will influence the image quality. The definition of "attractive" and "unattractive" natural images should be more reasonable and given.
2. The participants in this study are aged between 19 and 25. I believe that the influence of viewing nature images on emotions differs for different age groups. How do the authors look at the age differences?
3. For the AFFDEX tool for automated facial expression analysis, more details should be added to help readers understand how it works and if the performance is reliable. I would suggest the authors show some examples of the facial expression extraction process and results.
4. For the evaluation results, the analysis on 20 images (each category with 5 images) is quite limited to answering whether attractive natural scenes may evoke more positive emotions than unattractive scenes and whether self-reported emotions may be linked with objective measures of emotional facial expression. It's hard to know if and how the selection of images will influence facial emotion analysis.
Author Response
Dear reviewer,
thank you for your comments and recommendation. We tried to address all of them. Bellow, we described our changes and modifications. Your comments are in red color, our answers are black.
- This study investigated the effect of viewing attractive or unattractive natural scenes on emotional reactions. From the examples of the stimulus material shown in Figure 1, we can see that the "attractive" and "unattractive" are relative. The authors explained that "photos of unattractive scenes were taken on normal summer sunny days without any further adjustment". I don't think the adjustment is the main factor to distinguish "attractive" images from "unattractive" ones. For example, the nature scene, weather, camera angle, and even the image size will influence the image quality. The definition of "attractive" and "unattractive" natural images should be more reasonable and given.
We added broader explanation of our choice between attractive and unattractive images, see lines 199 -204. We use the same procedure as was utilized in in Joy and Bolderdijk's study [49]. It was based on the expert evaluation because it is supposed that experts can make that distinction on the basis of defined physical characteristics. As well as in that study, our open images are represented by of grand and dramatic mountain scenes, while unattractive images were taken in a mild Central-Bohemian landscape.
Of course, we did not claim that digital adjustment alone makes the image to be attractive. For instance, I can take image in forest after my home in normal, everyday atmospheric conditions that shows the forest scene as it really looks like. Or I can try to create attractive and mysterious photographs from this forest scene – I will wait for specific atmospheric condition (e.g., fog that can modulate sun rays), specific illumination - sunset or sunrise, etc. Moreover, with help of digital modification I can make from normal everyday scenery mysterious and attractive image. This reflects the difference between image (b) and (d) on Figure 1.
Of course, attractivity of environment may be influenced by culture. Maybe that for people living in high mountains might be attractive a flat arcadian agricultural landscape, and vice versa. For instance, according to my personal experiences people from Japan are exited while viewing our 19th century housing building that are boring (or even ugly) for us, but, in contrast, we are excited by common street from a Japanese town. We tried to explain (lines 215-220) that a possible effect of cultural differences does not play a role in our study, because our sample was culturally homogenous. Of course, cultural differences in similar type of research should consider, namely in countries, where population (and also in experimental sample) are not homogenous.
But we think that an effect of attractivity is not the main message of our paper. Because in our previous study we found very low magnitudes of facial emotional expressions while viewing natural images taken in the Central-Bohemian landscape, the next step in the current study was to employ different types of landscapes and, because they are different, to select them into different categories. While self-reported emotions (see 3.2. Analysis of self-reported emotions) clearly reflect our division between attractive/unattractive and open/closed images, we hope that our expert evaluation fit the participants' presentations of these visual categories.
- The participants in this study are aged between 19 and 25. I believe that the influence of viewing nature images on emotions differs for different age groups. How do the authors look at the age differences?
This comment is worthy of consideration; however, we do not think that of viewing nature images on emotions could systematically differs among different age groups.
In general, there is no evidence for any age effect in preference for natural scenes in environmental psychology literature (e.g., R.S. Ulrich, Aesthetic and affective response to natural environment I. Altman, J.F. Wohlwill (Eds.), Human Behavior and Environment, vol. 6, Plenum Press, New York (1983), pp. 85-125; R.S. Ulrich, Human responses to vegetation and landscapes, Landscape and Urban Planning, 13 (1986), pp. 29-44).
Theoretically, it is based on evolutionary hypotheses -“love for nature is in our genes” (e.g.,J. Appleton, The Experience of Landscape Wiley, London (1975); J.D. Balling, J.H. Falk, Development of visual preference for natural environments, Environment and Behavior, 14 (1982), pp. 5-28; J.H. Falk, J.D. Balling,Evolutionary influence on human landscape preference. Environment and Behavior, 42 (2010), pp. 479-493; S.R. Kellert, E.O. Wilson (Eds.), The Biophilia Hypothesis, Island Press, Washington).
Clearly, features of natural environments are constant over generations and are not affected by fashion trends or social, economic, and political changes, etc. Of course, older people, which are not in good physical conditions, would not prefer their free time relaxation climbing high mountains, but it does not mean that they would react on images with wild nature in a different way that you people.
- For the AFFDEX tool for automated facial expression analysis, more details should be added to help readers understand how it works and if the performance is reliable. I would suggest the authors show some examples of the facial expression extraction process and results.
We inserted the subsection “1.3.1. AFFDEX software for automatic computer facial expression analysis“, where we described, how the software works (lines 115-141).
- For the evaluation results, the analysis on 20 images (each category with 5 images) is quite limited to answering whether attractive natural scenes may evoke more positive emotions than unattractive scenes and whether self-reported emotions may be linked with objective measures of emotional facial expression. It's hard to know if and how the selection of images will influence facial emotion analysis.
We discussed this problem as the limitation of the research on lines ….. „Another limitation is the small number of photographs that were used in the study, as well as their specific selection. Obviously, natural scenes take many diverse forms all over the world, and a single investigation cannot include them all. Thus, the present study does not claim to be generalizable. Further replication studies are needed.”
Moreover, from practical reasons, to have more stimuli in this type of experiment may influence facial expressions – people could be after a longer time simply bored and do not express any facial emotions. To generalize these findings further replication studies are needed.
We also stressed exploratory character of the current study in the goals. Because the lack of findings in this field it is useful to have some data, even if their generalization is rather limited.
Round 2
Reviewer 1 Report
I think the manuscript has been quite well improved in all its section.
I’m still not sure about the implemented statistical analysis.
I understand your concern about the MANOVA, but I think an overall ANOVA model with different level of the “emotion” and of the “involvement” predictors could be the correct solution. In other words, I think it is a statistically wrong procedure to perform analysis on each “level”/category separately because of the experiment-wise error.
Please, clarify why not to use a on overall mixed between-within ANOVA model for both your analyses.
Author Response
We removed the calculation of one-way repeated measures ANOVA (lines 320-327). Originally, our intention was to confirm that emotion Joy reached higher magnitudes than other emotions. But we agree with you that this approach was not correct, because we omitted the other factors influencing the variable in this analysis.
Thank you again for your comments and recommendations that improved our manuscript.
Reviewer 2 Report
I have read the revision and responses, the authors addressed my previous comments.
Author Response
Thank you very much for your effort and constructive comments.